# HIV-1 Subtype Diversity in Morocco: Signals of Change and Implications for National Surveillance

**DOI:** 10.3390/idr17050128

**Published:** 2025-10-14

**Authors:** Maryam Ahmina, Hicham El Annaz, Nada Lamrak, Ahmed Reggad, Mohamed Rida Tagajdid, Rachid Abi, Mohamed Elqatni, Abdelilah Laraqui, Safae Elkochri, Elarbi Bouaiti, Youssef Aadi, Bouchra El Mchichi, Nadia Touil, Khalid Ennibi, Idriss Lahlou Amine

**Affiliations:** 1Molecular Virology and Onco-Biology Research Team, Faculty of Medicine and Pharmacy of Rabat, Rabat 10100, Morocco; maryam_ahmina@um5.ac.ma (M.A.);; 2Faculty of Medicine and Pharmacy, University Mohammed V in Rabat, Rabat 10100, Morocco; 3Center of Virology, Infectious and Tropical Diseases, Mohamed V Military Instruction Hospital, Rabat 10045, Morocco

**Keywords:** HIV-1, molecular epidemiology, Morocco, subtypes, molecular surveillance

## Abstract

Background: Limited molecular surveillance continues to constrain Morocco’s HIV response, leaving subtype dynamics largely underreported. Once characterized by a predominance of subtype B, the Moroccan epidemic now appears to reflect shifting patterns shaped by regional and international connectivity. This study aimed to investigate HIV-1 molecular diversity, monitor circulating HIV-1 genetic variants, and inter-gene recombination in a cohort of people living with HIV in Morocco. Methods: We conducted an analysis of individuals diagnosed with HIV-1 infection or receiving follow-up care. Demographic and clinical data were extracted. Genotypic testing was performed on the protease/reverse transcriptase (PR/RT) and integrase (IN) regions of the pol gene using the HIV-1 Genotyping Kit with Integrase. Subtypes were assigned via Stanford HIVdb and HIV Blast, and phylogenetic relationships were analyzed using MEGA 12. Results: Of the 73 individuals enrolled, 64 were successfully sequenced. The median age was 43 years (IQR 35–51.3), with over half aged 25–44, and 85.9% were male. Heterosexual transmission was the main route (87.5%), and 59.4% were ART-naïve. Non-B subtypes predominated (87.5%), led by CRF02_AG (73.4%), followed by B (12.5%), C (7.8%), and A3 (3.1%). The cohort showed significant genetic diversity, including multiple CRFs such as CRF45_cpx (1.6%), CRF01_AE (1.6%), B/CRF02_AG (7.8%), G/CRF02_AG (3.1%), C/CRF02_AG (1.6%), CRF02_AG/CRF45_cpx (1.6%) and CRF02_AG/CRF22_01A1 (1.6%). Conclusions: This study provides updated insight into HIV-1 diversity in Morocco, showing a predominance of non-B subtypes, particularly CRF02_AG, and signals of increasing heterogeneity compared with reports from more than a decade ago that described subtype B predominance. These findings suggest a viral transition shaped in part by regional connectivity and highlight a gap in Morocco’s HIV strategy, underscoring the need to implement nationwide molecular surveillance to inform future HIV control efforts.

## 1. Introduction

The global HIV-1 epidemic continues to evolve, not only in its geographic spread but also in its genetic complexity [1]. Nearly 40 million people are currently living with the virus and over 630,000 HIV-related deaths were recorded in 2024 [2,3]. HIV-1 is distinguished by its high genetic diversity, consisting of four groups (M, N, O, and P), with Group M alone accountable for the pandemic [4]. Group M is further classified into multiple subtypes (A–D, F–H, J, K) and more than 150 circulating recombinant forms (CRFs) and unique recombinant forms (URFs) arising from co-infection and recombination events [5,6,7,8,9,10]. While subtype B accounts for less than 15% of global infections, it remains the most studied due to its prevalence in high-income countries. In contrast, non-B subtypes such as CRF02_AG, A1, and C now dominate in regions with the highest HIV burden [6,8,11,12,13,14,15,16]. HIV-1 genetic diversity results from rapid nucleotide substitution and frequent recombination [4,17,18]. This heterogeneity influences viral virulence, disease progression, and most importantly the emergence of antiretroviral (ARV) resistance. Variability in the *env* gene enables immune evasion, while evolution in the *pol gene*, particularly in *reverse transcriptase, protease, and integrase*, drives resistance mutations that threaten sustained viral suppression [17,19,20,21,22]. Beyond resistance, this heterogeneity remains a major barrier to effective vaccine development, making molecular surveillance a cornerstone of effective public health strategies [23].

By the end of 2024, an estimated 23,500 people were living with HIV in Morocco, with a national prevalence of 0.08% and around 1000 new infections annually. Morocco has made significant advancement in its HIV response, with 80% of individuals living with HIV aware of their status, 95% receiving antiretroviral therapy (ART), and 95% achieving viral suppression, thereby moving closer to the UNAIDS 95-95-95 targets [24,25]. Recent initiatives to improve genotyping accessibility illustrate the nation’s growing commitment to incorporating molecular tools into its national strategy yet contemporary molecular data remain sparse and largely derived from clinically selected resistance-testing cohorts rather than systematic surveillance. Historic reports described subtype B predominance, with later observations of increasing CRF02_AG and other non-B lineages, but most of these data are more than a decade old [26,27,28,29,30,31,32]. Given Morocco’s position between sub-Saharan Africa, North Africa, and Europe, ongoing introductions of non-B variants through regional mobility are plausible and underline the need for updated, programme-feasible molecular monitoring to support treatment outcomes, drug resistance management, and epidemic control [12].

In this context, we conducted a molecular epidemiology study of HIV-1 in a Moroccan cohort newly diagnosed with HIV-1 infection and/or receiving routine follow-up care. Using genotypic sequence data from *pol* regions (PR/RT and IN), we evaluated subtype distribution and identified circulating recombinant forms (CRFs). This study provides an updated molecular snapshot of HIV-1 diversity in a Moroccon referral hospital, addressing gaps in national surveillance and underscoring the need to integrate molecular surveillance into the country’s HIV strategy.

## 2. Materials and Methods

### 2.1. Study Population

The study population comprised two groups: ART-naïve individuals, newly diagnosed, with baseline plasma viral load greater than 4 log_10_ copies per mL, and ART-experienced patients in routine follow-up who were referred by their clinicians for genotypic resistance testing owing to persistent viremia above 4 log_10_ copies per mL, consistent with virological failure. between August 2024 and May 2025 at the Center of Virology, Infectious and Tropical Diseases, Mohamed V Military Instruction Hospital, Rabat Morocco. Patients were eligible for sequencing if they had sufficient plasma viral load to allow amplification of the PR/RT and integrase regions of the pol gene. Individuals under the age of 18 were excluded. Clinical, demographic, and laboratory data were extracted from medical records. The study was conducted following the principles of the Declaration of Helsinki and was approved by the Ethics Committee of the Faculty of Medicine and Pharmacy at Mohammed V University, Rabat, Morocco (Approval ID: CERB 47/23). Written informed consent was obtained from all participants. All phylogenetic and statistical analyses were performed using de-identified datasets to ensure patient confidentiality.

### 2.2. Genotypic Study and Phylogenetic Analysis

HIV-1 viral load quantification was performed using the COBAS^®^ 4800 System (Roche Diagnostics, Mannheim, Germany) with a detection limit of 20 copies/mL. CD4+ T-cell counts were measured on a Beckman Coulter NAVIOS flow cytometer (Beckman Coulter Life Sciences, Brea, CA, USA). Genotypic testing was performed retrospectively using stored samples from individuals with ongoing HIV replication. Viral RNA was extracted from 400 μL blood plasma samples and eluted in 60 μL, using Qiagen EZ1 Advanced XL for Nucleic Acid Purification (Qiagen, Hilden, Germany). The viral RNA was used immediately for reverse transcription polymerase chain reaction (RT-PCR), followed by a nested PCR of the protease (PR), reverse transcriptase (RT), and integrase (IN) of the pol gene using the HIV-1 Genotyping Kit with Integrase (Applied Biosystems, Thermo Fisher, Carlsbad, CA, USA) in a GeneAmp PCR System 9700 thermal cycler (Applied Biosystems, Carlsbad, CA, USA). Sequencing was performed on a 3500XL Genetic Analyzer (Applied Biosystems Instruments, Foster City, CA, USA) using POP-7 polymer and capillary electrophoresis. Nucleotide sequences were manually inspected for quality, and ambiguities were resolved by reviewing chromatogram peak profiles. Raw sequence chromatograms were assembled and edited using SeqScape Software version 3.0 (Applied Biosystems, Foster City, CA, USA). Consensus sequences were generated and aligned for subtype analysis.

HIV-1 subtypes were assigned in two steps. First, sequences were analyzed with the Stanford University HIV Database (HIVdb) subtyping module (version 9.8) to obtain preliminary classifications. These results were then confirmed using the HIV BLAST tool version 2.13.0 against curated reference sequences from the Los Alamos National Laboratory (LANL) HIV database (https://www.hiv.lanl.gov/content/sequence/BASIC_BLAST/basic_blast.html (accessed on 23 September 2025)). For phylogenetic reconstruction, one representative reference sequence was selected from LANL for each subtype identified in our cohort, ensuring comprehensive representation while avoiding redundancy. Phylogenetic trees were inferred using the neighbor-joining method based on the Kimura 2-parameter model, with 1000 bootstrap replicates, in MEGA version 12. Bootstrap values ≥ 70 were considered to indicate reliable clustering.

### 2.3. Statistical Analysis

Categorical variables were summarized as frequencies and percentages and compared using Pearson’s χ^2^ test or Fisher’s exact test, as appropriate. Continuous variables were reported as medians with interquartile ranges (IQR1-IQR3) and compared using the non-parametric Mann–Whitney U test. All tests were two-sided, and a *p*-value of <0.05 was considered statistically significant. Statistical analyses were performed using SPSS Statistics, version 25.0 (IBM Corp., Armonk, NY, USA).

## 3. Results

### 3.1. Cohort Characteristics

Of the 73 individuals initially enrolled, 64 were successfully sequenced and included in the analysis. The remaining 9 samples could not be amplified, due to a combination of suboptimal plasma quality, RNA degradation during storage, and technical amplification failure despite repeated attempts. The demographic and clinical characteristics of the patients included in the study are summarized in Table 1. Overall, 8 patients (12.5%) were infected with HIV-1 subtype B and 56 (87.5%) with non-B subtypes, including CRFs. The median age was 43 years (IQR 35–51.3), with no significant difference between subtype B (43.5 years; IQR 36.5–47.3) and non-B (43 years; IQR 34.8–51.3; *p* = 0.750). Patients aged 25–44 years represented the most affected group (56.3%), followed by the patients aged >50 years (26.6%). The age category distribution did not differ significantly between subtypes (*p* = 0.777). The cohort was predominantly male (85.9%). No significant sex differences were noted between groups (*p* = 0.892). Regarding marital status, 57.8% have been married at some point, while 42.2% remain unmarried (*p* = 0.891). Heterosexual contact was the main transmission route (96.8%), with only two isolated cases of MSM and bisexual transmission (1.6% each); no significant variation by subtype was observed (*p* = 0.944).

At the time of diagnosis, 43.7% of individuals were classified as CDC stage A, 21.9% as stage B, and 34.4% as stage C. The difference between the groups was not statistically significant (*p* = 0.255). ART-naïve patients constituted 59.4% of the cohort, whereas ART-experienced patients comprised 40.6% and exhibited a higher likelihood of being infected with subtype B (*p* = 0.253). The median plasma viral load at the time of genotyping was 5.10 log_10_ copies/mL (IQR 4.51–5.70), with comparable levels in subtype B (5.10; IQR 4.09–5.70) and non-B (5.19; IQR 4.60–5.70; *p* = 0.280). Median CD4 count was 202 cells/mm^3^ (IQR 85–354), lower in subtype B infections (105; IQR 81–273) than in non-B (214; IQR 92–363).

### 3.2. Phylogenetic Analysis & Subtype Distribution

HIV-1 subtype analysis was successfully performed in 64/73 participants. The sequences were then analyzed to determine HIV-1 subtype distribution in the cohort. To further investigate the molecular dynamics of HIV-1 in our study, a phylogenetic analysis was conducted based on aligned PR/RT and IN sequences. A detailed description is provided in Figure 1. CRF02_AG was by far the most prevalent variant, identified in 47/64 participants (73.4%), of whom 57.8% (37/47) harbored pure CRF02_AG across both PR/RT and IN regions, while the remaining ten sequences showed discordant subtype assignments between the PR/RT and integrase regions of the pol gene, suggestive of inter-subtype recombination involving CRF02_AG in combination with other subtypes, including B (7.8%), G (3.1%), C (1.6%), CRF45_cpx (1.6%), and CRF22_01A1 (1.6%). The second common subtype was B, observed in 12.5% (8/64), followed by subtype C with 7.8% (5/64) and subtype A3 with 3.1% (2/64). The cohort also exhibited remarkable genetic diversity with multiple CRFs, including CRF01_AE and CRF45_cpx (1.6% each) (Figure 2).

## 4. Discussion

The epidemiological characteristics of the cohort reflect broader national trends and longstanding gaps in the HIV care cascade. In our cohort, men represented the majority of cases (85.9%). This male predominance reflects the nature of the study setting at the Military Training Hospital Mohammed V in Rabat, where many patients are members of the armed forces. Although relatives and civilians are also treated at CVMIT, this sex imbalance contrasts with national data, where women account for approximately 47% of all people diagnosed with HIV in Morocco, reinforcing the feminization of the Moroccan epidemic. This disparity may reflect differences in testing practices, healthcare access, or late diagnosis among women and highlights the need for gender-responsive strategies in HIV screening and care. More than half of our cohort patients were in the 24–44 age category. This age distribution is consistent with national surveillance reports, indicating that individuals aged 25 to 44 remain the most affected in Morocco across both sexes [25]. Heterosexual transmission remained the dominant mode of acquisition (87.5%), a proportion that aligns closely with previous Moroccan reports over the past two decades. Between 2004 and 2015, heterosexual transmission consistently represented the primary route of infection, ranging from 81.25% to 92.3% across multiple cohorts. This epidemiological profile reflects the spread of HIV beyond high-risk groups and into the broader population [26,28,29,30,33]. Data on men who have sex with men (MSM) remain limited in Morocco, although studies are currently underway to help address this critical gap [34]. Existing evidence indicates ongoing HIV transmission within this population; national surveillance estimates reported a prevalence of 4.2% in 2012, which increased to 5.3% by 2023 [33,35,36]. In our cohort, 57.8% of patients were married or had previously been married, while 42.2% were unmarried. In Morocco, where marriage is institutionally and socially defined as heterosexual, marital status remains an epidemiologically relevant demographic variable. Although it cannot fully capture non-marital partnerships or unreported same-sex practices, integrating marital status into molecular surveillance may provide useful insights into transmission dynamics within couples or stable relationships and inform the design of more targeted interventions [37].

Our cohort consisted of patients referred for genotypic testing due to persistent viral replication, a selection criterion that inherently enriched the study population with ART-naïve individuals and ART-experienced patients undergoing virologic failure. The ART coverage distribution in our study does not align with national programmatic data, which indicates that over 95% of individuals diagnosed with HIV are receiving treatment, and most attain viral suppression. Our findings identify a subgroup of patients at risk for adverse outcomes, revealing significant deficiencies in the care continuum that molecular surveillance can effectively monitor.

Our cohort showed a median CD4 count of 202 cells/mm^3^ and a median viral load of 5.19 log_10_ copies/mL. In contrast, over the past two decades in Morocco, median CD4 counts have gradually increased from 116 cells/mm^3^ in 2005–2009 to over 400 cells/mm^3^ in cohorts from 2014–2015 in ART-naïve patients [28,29] and from 346 cells/mm^3^ in 2005–2010 to over 400 cells/mm^3^ in cohorts from 2014–2015 in ART-experienced patients [26,27], while median viral loads have decreased from approximately 5.17 log_10_ copies/mL to 4.56 log_10_, suggesting that despite national progress, late diagnosis and immune suppression remain common [26,28,30,31]. These immunological and virologic findings are consistent with the CDC stage distribution in our cohort, where 34.4% of patients were classified as stage C, confirming that a considerable proportion of individuals were diagnosed at an advanced stage of HIV infection. In contrast, 43.7% were diagnosed at an asymptomatic stage (stage A), which aligns with national surveillance data showing increased early detection in recent years. Despite substantial advancements in testing expansion and enhanced access to ART, our findings highlight that late diagnosis continues to be a persistent public health challenge [25].

Our study suggests a shift in the molecular epidemiology of HIV-1 in Morocco, with a predominance of non-B subtypes (87.5%) and increasing viral diversity observed within the cohort. In earlier molecular epidemiology studies, HIV-1 subtype B was overwhelmingly predominant in Morocco, representing 93.5% of infections in 1997, while subtypes A and F accounted for just 1.0% and 0.5%, respectively [38]. By 2005, although subtype B remained the most common (76.7%), there was a marked rise in non-B subtypes, particularly CRF02_AG, which constituted 15% of cases [32]. Between 2004 and 2015, subtype B remained dominant across most molecular studies, with proportions ranging from 66% to 74% [26,28,30]. However, during the same period, a gradual increase in non-B subtypes—particularly CRF02_AG—was documented, rising from 9% in 2005–2010 to 25.3% by 2006–2010 [26,30]. The appearance of other subtypes, such as A1, C, and unique recombinants, has also been intermittently reported [26,28,30,31]. Our current findings demonstrate a continued and substantial shift in subtype dynamics, with CRF02_AG now accounting for 73.4% of cases and subtype B declining to just 12.5%, reflecting an ongoing diversification of circulating strains. These results underscore a clear epidemiological transition over the past two decades and highlight the increasing complexity of HIV molecular patterns in Morocco.

Morocco’s strategic location at the crossroads of Africa and Europe may contribute to its exposure to diverse HIV-1 variants. The predominance of CRF02_AG (73.4%) in our study aligns with global trends reported in West and Central Africa [14,39,40,41]; with Cameroon at 60–68% in 2016 [42], where this recombinant form is highly prevalent, unlike much of the MENA region, where subtype B remained the most prevalent (39%) until 2016 in countries like Algeria, Tunisia, and Yemen [43,44]. The persistence of subtype B in our cohort suggests a historical link to earlier European transmission networks where subtype B is still the most widespread viral strain, especially in countries like France with 56%, Germany and Spain, which exceeded 80% [45,46]. As Europe now sees an influx of non-B subtypes due to migration and with globalized transmission, other subtypes have also been introduced; for example, in Spain, subtype F1 was identified in a cohort of MSM and CRF02_AG among immigrant populations [46,47,48,49]. Our study also identified several other subtypes and recombinants, including subtype C, typically associated with Southern and Eastern Africa [50], subtype G, the second most prevalent in West Africa [51], and complex recombinant forms such as CRF04_cpx, originating in Central Africa; CRF45_cpx, previously reported in Cameroon [7,14] and CRF01_AE, common in Southeast Asia [52,53]. Importantly, HIV prevalence among migrants in Morocco has been reported as markedly higher (4.6%) than in the general population (<0.1%), and this group faces significant barriers to accessing healthcare [25,54,55]. Although our study did not include patient-level migration data, these contextual data support the hypothesis that Morocco’s geographic position at the crossroads of Africa and Europe may facilitate exposure to globally circulating variants. Morocco appears to be experiencing a parallel diversification, acting as both a recipient and conduit for recombinant HIV-1 forms. While our data cannot directly establish the drivers of this diversification, these findings are consistent with Morocco’s growing exposure to globally circulating variants and reinforce its role as a molecular crossroads in the international HIV epidemic.

An important observation in our study was the variation in subtype classification between the PR/RT and integrase (IN) regions in 10 patients, highlighting subtype discordance. The most frequent pattern was subtype B in PR/RT and CRF02_AG in IN, although other combinations involving CRF02_AG with subtypes C, G, CRF45_cpx, and CRF22_01A1 were also detected. These inter-gene discrepancies are consistent with possible recombination events, a phenomenon well documented in high-diversity settings where co-infection or superinfection facilitates template switching during reverse transcription [18].

The growing complexity of HIV-1 diversity in Morocco underscores the need to strengthen surveillance strategies. At present, genotypic resistance testing is not routinely performed at diagnosis and is generally limited to cases where it is specifically requested or conducted within research settings. By including ART-naïve patients, our study provides baseline molecular data that suggest it may be timely to consider the implementation of baseline resistance testing for all new diagnoses as part of Morocco’s HIV response.

Our findings should be interpreted considering several limitations. First, this was a single-center observational study with a modest sample size, which may limit the generalizability of the results. The study population consisted of individuals engaged in clinical care and eligible for genotypic testing, introducing the potential for selection bias. Second, the cohort was predominantly male, reflecting the military setting of the hospital. Third, sequencing was restricted to the pol gene, which prevents definitive characterization of recombinant viruses and may underestimate the true extent of genetic diversity. Finally, our study provides only a snapshot of circulating strains without capturing longitudinal changes. Nevertheless, the study population included not only military personnel but also their relatives and a substantial number of civilian patients referred from different regions of Morocco, offering a valuable window into the molecular diversity observed in clinical practice.

## 5. Conclusions

This study provides one of the most up-to-date molecular snapshots of HIV-1 diversity in Morocco, filling a gap of more than a decade since the last published data, which had reported a predominance of subtype B. Our analysis shows that non-B subtypes, particularly CRF02_AG, now predominate, with indications of increasing heterogeneity within the epidemic. These findings provide signals of changing molecular epidemiology of HIV-1 in Morocco, though broader studies are needed to confirm national trends. Molecular surveillance should be urgently integrated into the national HIV strategy to anticipate subtype dynamics, strengthen public health interventions, and optimize treatment outcomes.

## Figures and Tables

**Figure 1 idr-17-00128-f001:**
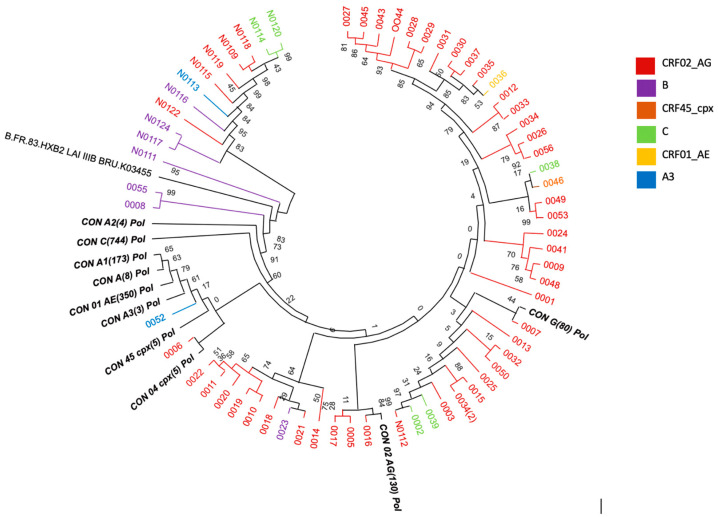
Neighbor-joining phylogenetic tree of HIV-1 subtype pol sequences using MEGA 12. Reference sequences representing major HIV-1 subtypes and circulating recombinant forms (CRFs) were included to determine subtype clustering. Bootstrap analysis was performed with 1000 replicates, and values ≥ 70 are displayed at branch nodes.

**Figure 2 idr-17-00128-f002:**
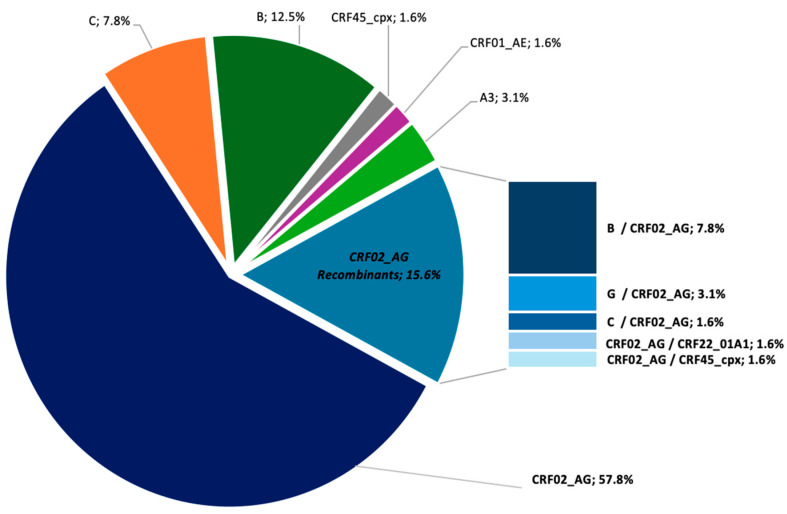
Distribution of HIV-1 Subtypes and CRFs 4.

**Table 1 idr-17-00128-t001:** Demographic, clinical, and immunovirological characteristics of the 64 HIV-1-infected patients at the time of sampling according to viral subtype (B vs. non-B).

	Total N = 64	Subtype B N = 8	Subtype Non-B N = 56	*p*-Value
Median age, years	43 (35–51.3)	43.5 (36.5–47.3)	43 (34.8–51.3)	0.750
Age Categories				0.946
19–24	2 (3.1%)	0	2 (3.1%)	
25–44	36 (56.3%)	5 (7.8%)	31 (48.4%)	
45–50	9 (14.1%)	1 (1.60%)	8 (12.5%)	
>50	17 (26.6%)	2 (3.10%)	15 (23.4%)	
Sex				0.892
Male	55 (85.9%)	7 (10.9%)	48 (75%)	
Female	9 (14.1%)	1 (1.60%)	8 (12.5%)	
Marital Status				0.891
Unmarried	27 (42.2%)	4 (6.30%)	23 (35.9%)	
Ever married	37 (57.8%)	4 (6.30%)	33 (51.6%)	
Route of transmission				0.944
Heterosexual	62 (96.8%)	8 (12.5%)	54 (84.3%)	
MSM	1 (1.6%)	0	1 (1.6%)	
Bisexual	1 (1.6%)	0	1 (1.6%)	
CDC Stage				0.255
A	28 (43.7%)	4 (6.2%)	24 (37.5%)	
B	14 (21.9%)	0	14 (21.9%)	
C	22 (34.4%)	4 (6.3%)	18 (28.1%)	
ART-Status				0.253
ART-naïve	38 (59.4%)	3 (4.70%)	35 (54.7%)	
ART-experienced	26 (40.6%)	5 (7.80%)	21 (32.8%)	
Median Plasma VL (log Copie/mL)	5.19 (4.51–5.7)	5.10 (4.09–5.62)	5.19 (4.60–5.70)	0.280
Median CD4 cell counts (Cells/mm^3^)	202 (85–354)	105 (81–273)	214 (92–363)	0.260

## Data Availability

Restrictions apply to the availability of these data. The datasets generated and analyzed during the current study are not yet publicly available, as the sequencing and analysis are still ongoing. Data will be deposited in GenBank upon completion of the study. Requests to access the datasets should be directed to Hicham El Annaz: hichamelannaz74@gmail.com.

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
