# Peer review of "HIV-1 Subtype Diversity in Morocco: Signals of Change and Implications for National Surveillance"

_2036-7449, 2025, doi:10.3390/idr17050128_

Round 1

Reviewer 1 Report

Comments and Suggestions for Authors

I have added some comments to the PDF for consideration.  Overall, this is a nice piece of work, but I don't understand why the resistance data from the Stanford analysis were not included.  Those data would significantly enhance the manuscript.

Author Response

Thank you for the constructive feedback on our manuscript. We have revised the paper accordingly and uploaded it.
  1. a point-by-point response to all comments, and
  2. a clean and a marked-up manuscript with all changes highlighted in yellow is joined to this response.
Comment 1
“I don't understand why the resistance data from the Stanford analysis were not included. Those data would significantly enhance the manuscript.”
Response 1:
We sincerely thank the reviewer for this suggestion. We agree that resistance data generated through the Stanford HIVdb analysis are highly valuable and would provide important clinical insights. Our decision not to include them in the current manuscript was intentional, as we are preparing a separate article dedicated to HIV-1 drug resistance mutations in Morocco, using the same dataset. This will allow us to present the resistance results in greater detail, including mutation frequencies, drug-class implications, and their potential impact on treatment strategies. In the present manuscript, we specifically chose to focus on HIV-1 subtype distribution and molecular diversity, as our analysis revealed a clear shift from historical subtype B predominance toward non-B subtypes, particularly CRF02_AG. Including a detailed resistance analysis here would have risked diluting this central message.
We hope the reviewer understands this approach, which was designed to ensure clarity of scope in the current work while reserving resistance findings for a dedicated publication with the depth they merit.
Comment 2
“I would like to know more about the hospital in terms of how representative the patient population would be, and how generalizable the findings.”
Response 2:
We thank the reviewer for this pertinent question. The Center of Virology, Infectious and Tropical Diseases, Mohamed V Military Instruction Hospital, Rabat is a national referral center for HIV care in Morocco. Although it is a military hospital, CVMIT serves both military personnel and their families and a large number of patients, with referrals coming from multiple regions of the country. Therefore, the patient population extends beyond Rabat, encompassing individuals from a variety of geographic and social backgrounds.
Nevertheless, we acknowledge that our study cohort cannot be considered fully representative of all people living with HIV in Morocco. Our participants were individuals undergoing genotypic testing because of ongoing viral replication in the study period. Therefore, while our findings provide an important molecular snapshot of circulating variants, they should be interpreted as reflective of patients engaged in care at a referral center rather than the entire Moroccan HIV population.
We have added clarifications in the Methods and Discussion sections of the revised manuscript to highlight both the strengths (national referral scope) and the limitations (single-center, selective genotyping criteria) of our study population in terms of generalizability.
Comment 3
“What was the reason for the 9 cases that could not be sequenced?”
Response 3:
We thank the reviewer for this question. Of the 73 samples analyzed, 9 could not be successfully sequenced. The primary reasons were low plasma viral load close to or below the assay amplification threshold and RNA degradation in some stored samples, which resulted in unsuccessful PCR amplification despite repeated attempts.
We have now clarified the matter in the Methods section and added a note in the Results to explain the discrepancy between enrolled and successfully sequenced cases.
Comment 4
“Mediane is spelled incorrectly.”
Response 4:
We thank the reviewer for noticing this typographical error. We have corrected “Mediane” to “Median” in Table 1 and throughout the manuscript.
Comment 5
“In the absence of whole genome sequencing, I'm not sure that we can be confident these represent recombinants.”
Response 5:
We appreciate the reviewer’s important observation. In our cohort, 10 cases showed discordant subtype assignments between the PR/RT and integrase regions of the pol gene. We have revised the text to clarify that these findings are suggestive of inter-subtype recombination but should be interpreted with caution, given the limitations of partial genome sequencing.
Accordingly, we no longer describe these viruses as confirmed recombinants but rather as cases with discordant pol region profiles. We also highlight in the discussion that near full-length sequencing will be essential in future studies to confirm and better characterize recombinant forms in Morocco.
Comment 6
“This male predominance contrasts with national data… suggesting that the findings of this study might not be applicable to the country as a whole, as I suggested earlier.”
Response 6:
We thank the reviewer for this important observation. We agree that the predominance of male participants in our cohort contrasts with national HIV surveillance data, where women represent nearly half of people living with HIV in Morocco. This imbalance reflects the institutional setting of our study, as the Hôpital Militaire d’Instruction Mohammed V primarily treats military personnel, who are mostly men, although their relatives and civilian patients are also included in care.
We have revised the discussion to clarify this point and to emphasize that our findings should be interpreted as a molecular snapshot of patients receiving care at this referral center, rather than as a nationally representative sample.
Comment 7
“Other limitations include the predominantly male cohort.”
Response 7:
We thank the reviewer for this useful suggestion. We have now explicitly added this point to the Limitations section of the Discussion.
Comment 8
“In the conclusion, the authors mention a rapid evolutionary rate. I think this is an incorrect term: evolutionary rate of the virus has not been studied.”
Response 8:
We thank the reviewer for this clarification. We have therefore removed this term from the conclusion. Instead, we now describe our findings as reflecting increasing subtype diversity within the cohort, consistent with broader regional trends. This wording more accurately represents our data without implying an analysis of molecular clock or substitution rates.
Comment 9
“In the conclusion, the authors write ‘evolving virologic landscape.’ Better terminology and more accurate wording is needed.”
Response 9:
We thank the reviewer for pointing this out. We have revised the wording in the conclusion to more accurately reflect our findings.

Reviewer 2 Report

Comments and Suggestions for Authors

The manuscript is simple, revealing HIV variants in a small sample size, in patients who attended an antiretroviral resistance test. I agree with the author's emphasis on epidemiological surveillance and identifying potential changes in HIV subtypes within each country. However, I disagree with their conclusions and assertions regarding the methodology used to define viral variants, ensuring the change in HIV epidemiology in Morocco, in addition to the sample size and other limitations. I would recommend considering publication as a "brief report" rather than an original article.

Subtyping analysis is performed on regions of the viral genome related to ARV resistance of protease, reverse transcriptase, and integrase. For me, this is limiting, in order to determine the subtype of the virus, as they mention, the introduction is more complex. Furthermore, if they only want to analyze these regions to determine the virus subtype, they should mention this in the title. Hence, the reader understands the direction the article is going.

The conclusions made by the authors are very strong, mentioning that there is an evident change in subtype in Morocco. However, the sample size they used is small and not representative (64 individuals, 0.3% of the estimated population infected with HIV in Morocco, according to their data). Furthermore, the region and size of the viral genome sequenced are not appropriate for their assertion.

The authors describe demographic and clinical data, but do not relate them to the increase or change of HIV subtypes in Morocco. In several sections, they mention that the change in the viral population with subtype variation is given by the constant migration, for example, from Africa and Europe.

That conjecture is supported in the discussion; they do not have the necessary elements to do so. The authors did not investigate the patients' medical history regarding their relationship with other countries.

Their main conclusion is that they reveal a change in the molecular epidemiology of HIV-1 in Morocco. They mention that subtype B is no longer predominant. They only perform the analysis in a region of HIV where ARVs act, the protease, reverse transcriptase, and integrase. This should also be mentioned in the title. They need to explain their phylogenetic analysis, reference sequences, etc., more. If this is the basis of the work, they do not explain the groupings and background information that they might have about the reference sequences to which their samples are grouped. This section on the change in viral epidemiology in Morocco is supported by five articles, two of which are self-citations, and three of which focus primarily on ARV resistance.

The discussion contradicts its introduction, stating that 80 are diagnosed, 95 are on treatment, and 95 are virally suppressed. However, the discussion mentions that women do not have adequate access to healthcare and are also diagnosed late, yet they represent 47% of infections in Morocco. If a high percentage of patients living with HIV are on antiretroviral treatment (95% according to its introduction), why is this not represented in the patient samples? Did the authors include a cohort with a larger number of drug-naive patients? Or in Morocco, do only drug-naive patients have access to antiretroviral resistance testing? The authors emphasize marital status; however, this does not guarantee that the patient is not reporting their sexual preferences, because they may or may not have previously been married and have homosexual or bisexual preferences.

The authors mention a relationship between patients on antiretroviral treatment and having subtype B. This assertion cannot be sustained. The authors mention that of 26 patients on antiretroviral treatment, 5 (19%) had subtype B and 21 (81%) were non-B. If a relationship existed, the percentages should be reversed.

Author Response

Thank you for the constructive feedback on our manuscript. We have revised the paper accordingly and uploaded it.
  1. a point-by-point response to all comments, and
  2. a clean and a marked-up manuscript with all changes highlighted in yellow is joined to this response.
Comment 1:

Subtyping analysis is performed on regions of the viral genome related to ARV resistance of protease, reverse transcriptase, and integrase. For me, this is limiting ; in order to determine the subtype of the virus, as they mention, the introduction is more complex. Furthermore, if they only want to analyze these regions to determine the virus subtype, they should mention this in the title. Hence, the reader understands the direction the article is going.

Response 1:
We thank the reviewer for raising this important point. We agree that near full-length genome sequencing offers the most comprehensive approach to subtype assignment. However, the pol gene (PR/RT and IN regions) remains widely used and internationally validated for HIV-1 subtyping in epidemiological and surveillance studies. This is because pol harbors sufficient phylogenetic signal to accurately distinguish between major subtypes and circulating recombinant forms, and it is the region most consistently sequenced in routine drug resistance testing by capillary sequencing and NGS. Several large-scale molecular epidemiology studies, including WHO-referenced databases and regional surveillance projects, rely primarily on the pol gene for subtype determination.In our setting, as a lower-middle-income country, resources for full-genome sequencing are not yet widely available. Therefore, sequencing the pol region provides a pragmatic and cost-effective compromise, enabling molecular surveillance to be integrated into routine patient management while still generating reliable subtype data.

To ensure transparency, we have explicitly clarified in the methods and discussion that subtyping was based on pol regions, from a Single-Center Cohort. This makes the scope clear to the readers while justifying the methodological choice in our context.

Comment 2/3:

The conclusions made by the authors are very strong, mentioning that there is an evident change in subtype in Morocco. However, the sample size they used is small and not representative (64 individuals, 0.3% of the estimated population infected with HIV in Morocco, according to their data). Furthermore, the region and size of the viral genome sequenced are not appropriate for their assertion.

Response 2/3:

We acknowledge this important limitation and have revised our conclusions accordingly. We no longer present our findings as definitive evidence of a nationwide shift in HIV subtype epidemiology. Instead, we describe our results as an “updated molecular snapshot” from a single-center cohort. We also emphasize in both the Discussion and Conclusion that larger, nationwide studies are needed to confirm these trends.

We have softened our language throughout the manuscript. This more cautious interpretation acknowledges the preliminary nature of our study while highlighting the importance of molecular surveillance.

Comment 4:

The authors describe demographic and clinical data, but do not relate them to the increase or change of HIV subtypes in Morocco. In several sections, they mention that the change in the viral population with subtype variation is given by the constant migration, for example, from Africa and Europe. That conjecture is supported in the discussion; they do not have the necessary elements to do so. The authors did not investigate the patients' medical history regarding their relationship with other countries.

Response 4:

We thank the reviewer for this observation. In our Results, we presented demographic and clinical characteristics by subtype (Table 1), but as noted, no statistically significant associations were observed between demographic or clinical variables and subtype distribution. In the discussion, we have clarified that these variables reflect broader national epidemiological trends but do not directly explain subtype shifts in our dataset. We now explicitly state this limitation in the revised manuscript, noting that larger, multicenter studies with more representative sampling will be required to explore potential associations between clinical/demographic factors and subtype distribution in Morocco.

We agree that our study did not include patient-level data on travel or migration history. In response, we have revised the Discussion to temper our interpretation and to clearly state that our comments on migration are based on published contextual data rather than direct cohort evidence. We have also added references highlighting the higher prevalence of HIV among migrants in Morocco (≈4.6% vs <0.1% in the general population) and their barriers to healthcare access, which have been linked to the introduction and circulation of diverse subtypes. This provides a more balanced explanation that situates our findings within the broader epidemiological context while remaining cautious about causal attribution.

Comment 5:

Their main conclusion is that they reveal a change in the molecular epidemiology of HIV-1 in Morocco. They mention that subtype B is no longer predominant. They only perform the analysis in a region of HIV where ARVs act, the protease, reverse transcriptase, and integrase. This should also be mentioned in the title.

They need to explain their phylogenetic analysis, reference sequences, etc., more. If this is the basis of the work, they do not explain the groupings and background information that they might have about the reference sequences to which their samples are grouped.

This section on the change in viral epidemiology in Morocco is supported by five articles, two of which are self-citations, and three of which focus primarily on ARV resistance.

Response 5:

We thank the reviewer for this constructive feedback. In response, we have revised the manuscript in several ways. First, we have softened our conclusions and now present our findings as an “updated molecular snapshot” of HIV-1 diversity from a single-center cohort, rather than definitive evidence of a nationwide shift. We also emphasize in the Discussion and Conclusion that broader, nationwide studies are required to validate these trends.

Second, we clarified in the Methods that subtype assignment was performed using PR/RT and IN regions of the pol gene, which—although not covering the full genome—are widely used and internationally validated for subtyping in molecular epidemiology and resistance surveillance. To ensure transparency, we now explicitly state that our analysis was limited to these regions and have acknowledged this in the Limitations.

Third, we expanded our description of the phylogenetic approach: subtype assignment was performed in two steps (Stanford HIVdb and HIV BLAST against LANL references), and one representative reference sequence per subtype was selected to avoid redundancy. Phylogenetic trees were reconstructed using the neighbor-joining method with the Kimura 2-parameter model and 1000 bootstrap replicates in MEGA v12, with bootstrap values ≥70 considered robust. These details improve transparency and reproducibility.

Finally, regarding references, we recognize that the number of Moroccan studies on HIV-1 molecular epidemiology is limited (total of 6). This scarcity is itself a reflection of the evidence gap in Morocco, where sequencing has historically focused on drug resistance rather than subtype surveillance. Although some cited studies were resistance-focused, they also reported subtype distributions and therefore provide important historical context. To strengthen the discussion, we have incorporated references from the broader MENA region and global analyses, situating our findings within an international framework.

Collectively, these revisions improve the precision, transparency, and contextualization of our work, while more cautiously framing our contribution to the limited body of molecular epidemiology data from Morocco.

Comment 6:

The discussion contradicts its introduction, stating that 80 are diagnosed, 95 are on treatment, and 95 are virally suppressed. However, the discussion mentions that women do not have adequate access to healthcare and are also diagnosed late, yet they represent 47% of infections in Morocco. If a high percentage of patients living with HIV are on antiretroviral treatment (95% according to its introduction), why is this not represented in the patient samples? Did the authors include a cohort with a larger number of drug-naive patients? Or in Morocco, do only drug-naive patients have access to antiretroviral resistance testing? The authors emphasize marital status; however, this does not guarantee that the patient is not reporting their sexual preferences, because they may or may not have previously been married and have homosexual or bisexual preferences.

Response 6:
We thank the reviewer for this important observation. The figures in the Introduction (80–95–95) reflect national programmatic data and the broader Moroccan HIV care cascade. In contrast, our study population was not a random sample of all people living with HIV but specifically those undergoing genotypic testing due to detectable viral replication. This clinical selection criterion naturally enriches our cohort for ART-naïve individuals (59.4% in our study) and ART-experienced patients with virological failure. Consequently, our sample is not expected to mirror the general distribution of ART coverage and suppression in Morocco.

We have clarified this distinction in the Discussion to avoid any impression of contradiction. The apparent divergence actually underscores the value of genotyping studies: while national statistics highlight successes in treatment scale-up, our cohort reveals that a substantial proportion of patients still present with late diagnosis, ongoing replication, or treatment failure. These patients represent critical gaps in the cascade of care that molecular surveillance can help to monitor.

Regarding women, our discussion aimed not to contradict national prevalence data but to emphasize gender disparities in access and outcomes. Although women represent nearly half of Morocco’s PLHIV population, they were underrepresented in our cohort (14.1%), consistent with late testing and barriers to care documented in national reports. We have rephrased this section to clarify that our findings reflect subgroup disparities within a national context of overall progress.

We agree that marital status does not directly capture sexual preferences or behaviors. However, in the Moroccan sociocultural and legal context, marriage is institutionally and socially defined as heterosexual, and thus remains a relevant epidemiological variable when describing patient demographics. While it cannot account for non-marital partnerships or unreported same-sex practices, reporting marital status provides useful context for understanding transmission dynamics within couples or stable relationships. We have revised the text in the Discussion to clarify both the value and the limitations of using marital status in this setting.

Comment 7:
The authors mention a relationship between patients on antiretroviral treatment and having subtype B. This assertion cannot be sustained. The authors mention that of 26 patients on antiretroviral treatment, 5 (19%) had subtype B and 21 (81%) were non-B. If a relationship existed, the percentages should be reversed.

Response 7:

We agree with the reviewer that the numbers in our study are too small to establish a meaningful association between ART status and HIV-1 subtype. The apparent overrepresentation of subtype B among ART-experienced patients (19%) is not statistically significant (p = 0.253) and may simply reflect historical patterns of the epidemic in Morocco, where subtype B predominated in earlier years when many of the ART-experienced patients in our cohort were likely first diagnosed. In contrast, the predominance of CRF02_AG and other non-B subtypes among ART-naïve individuals reflects the ongoing diversification of the epidemic in more recent years.

We have removed the claim of a direct association between ART status and subtype B in the revised manuscript to avoid over-interpretation. We also highlight this as an area for future investigation with larger datasets and longitudinal follow-up.

Comment 8:

I would recommend considering publication as a "brief report" rather than an original article.

Response 8:
We respectfully thank the reviewer for this suggestion. However, we believe our manuscript is best positioned as an original article rather than a brief report. To our knowledge, this is one of the few studies in Morocco to analyze both PR/RT and integrase regions, thereby offering a more comprehensive molecular snapshot than prior work. Data on HIV-1 genetic diversity and recombination in Morocco remain extremely scarce, with most available studies over a decade old and primarily limited to resistance-focused analyses. By including newly diagnosed ART-naïve patients as well as ART-experienced individuals with virological failure, our study provides novel insights into both circulating subtypes and the complexity of recombination patterns. In this context, we feel the scope, depth, and rarity of the data justify its consideration as an original article.

Round 2

Reviewer 2 Report

Comments and Suggestions for Authors

The authors made significant improvements to their work. In my opinion, I continue to recommend considering the publication as a "brief report" rather than an original article.

I think, as the authors mention, it is important to report this finding; however, the limitations of the study, which includes a very small sample and the number of results, are not sufficient for an original article.

I leave this suggestion to the editor for consideration.